# ACCELERATING VISION TRANSFORMERS WITH DROP-IN DEPTHWISE CONVOLUTION

## ABSTRACT

Pretrained vision foundation models deliver strong performance across tasks with limited fine-tuning. However, their Vision Transformer (ViT) backbones impose high inference costs, limiting deployment on resource-constrained devices. In this work, we accelerate large-scale pretrained ViTs while preserving their feature extraction capabilities by exploiting the intrinsic convolution-like behavior of some attention heads. Specifically, we introduce an efficient depthwise convolution-based layer that serves as a drop-in replacement for these heads. Additionally, we propose simple strategies to identify which heads can be replaced and introduce a fine-tuning procedure that recovers downstream task performance. Across both image classification and segmentation tasks, our method achieves 17–20% inference speedup with minimal performance degradation. We validate the approach through detailed derivations, extensive experiments, and efficiency benchmarks on multiple low-power platforms. Implementation will be released publicly.

Large-scale-pretrained Vision Transformers (ViTs) (Dosovitskiy et al. (2021)) have emerged as a powerful paradigm in modern machine learning. With minimal fine-tuning, they can perform competitively on a wide range of downstream vision tasks. Dominating paradigms include DINO (Caron et al. (2021); Oquab et al. (2023)), MAE (He et al. (2022)), and CLIP (Radford et al. (2021)). A noticeable downside is the high inference cost of the ViT architecture, especially when targeting inference on low-power edge devices. Although various methods have been proposed to reduce the inference cost of ViTs, several challenges remain. Popular approaches leverage a reduction in token counts (Bolya et al. (2023); Meng et al. (2022); Graham et al. (2021)), which disrupts the spatial structure of the features and limits the applicability to dense prediction tasks (i.e, segmentation). Another common occurrence is the reliance on dynamic conditional execution flow or dynamically shaped tensors, seldom supported by high-performance inference frameworks. In addition, for fine-tuned foundation models, it is fundamental to preserve the large-scale pretrained weights, which is usually not the case in efficient transformer backbones (Cai et al. (2023); Mehta & Rastegari (2022a); Wadekar & Chaurasia (2022)).

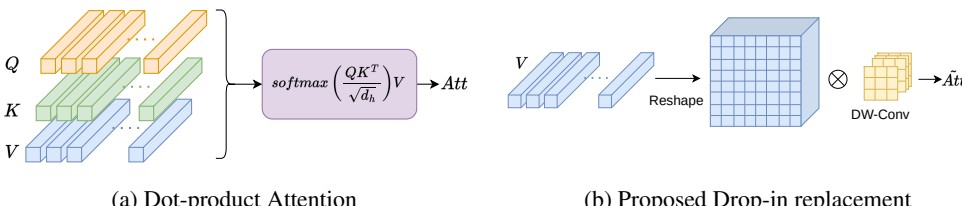

(a) Dot-product Attention    (b) Proposed Drop-in replacement

Figure 1: **Illustration of the proposed drop-in approximation.** We replace attention (a) with a Depthwise convolution (b), which improves inference speed while reusing the pre-trained network parameters for performance.

To address these challenges, in this paper, we propose an efficient, *drop-in acceleration method for foundation ViTs*. Building on previous research (Section 1.2), we assume that several Multi-head Self Attention (MhSA) Vaswani et al. (2017) heads learn highly localized, static patterns, with a structure closely resembling convolution. We propose to replace a carefully selected set of computationally intensive Self-Attention operators with a much more efficient depthwise convolution

over the reshaped Value tensors (Figure 1). Our method works as a drop-in replacement, recovering the performance of the large-scale pretrained model with limited fine-tuning, showing minimal performance loss while achieving 17% to over 20% speedup.

The contributions of this work can be summarized as follows.

- We derive an efficient formulation that serves as a drop-in replacement for attention heads that learn particular convolution-like structures. We later show that head ensembling from He et al. (2024) can be made explicit and generalized to our setting, benefiting from the more efficient formulation.

- We propose a simple methodology to identify the heads to be approximated by convolution. We also validate the proposed heuristic against a more sophisticated solution that we reformulate for this context.

- We explicitly consider the problem of performance optimization in the context of pretrained foundation models, targeting a realistic deployment scenario, focusing on a popular edge platform device (Nvidia Orin Nano) and an appropriate inference framework (TensorRT). We later extend the analysis to multiple specialized platforms.

We believe that the work presented addresses numerous challenges that have been only partially covered in the literature. It also allows for casting the performance improvement of vision foundation models from a different perspective, which in the future might become part of more advanced pruning frameworks.

# 1 RELATED WORK

## 1.1 PRUNING OF TRANSFORMERS

The idea of reducing the complexity of a neural network by eliminating less important parameters, connections, and layers dates back to the early deep learning era (LeCun et al. (1989); Hassibi & Stork (1992)). Similar techniques, while applicable to BERT Devlin et al. (2019) style transformers (Sanh et al. (2020); Chen et al. (2020); Brix et al. (2020)), induce *unstructured* sparsity, often causing overhead due to irregular memory access. Some works partially address this via block sparsity (Lagunas et al. (2021); Xu et al. (2024)). Structured pruning of attention heads was explored in Michel et al. (2019), with Voita et al. (2019) introducing stochastic gating to select heads during training. Building on this, Behnke & Heafield (2020) identifies prunable heads early using confidence scores, while DSP Li et al. (2021) introduces explicit control over pruning ratios.

Some pruning methods for ViTs require access to training (Prasetyo et al. (2023); Lin et al. (2024)), others rely on distillation (Yu et al. (2022); Yang et al. (2023)). Many focus on pretrained DeiT models (Touvron et al. (2021)) for classification (Zheng et al. (2022); Yu et al. (2022); He et al. (2024); Lin et al. (2024)). SPViT He et al. (2024) prune MhSA blocks during fine-tuning into learned convolutional layers formulated under the sufficient condition of Cordonnier et al. (2020). That assumption implicitly collapses the block into a single convolution. In contrast, we derive a more general formulation applicable to pretrained foundation models, and later show how SPViT arises as a special case within our framework. Lambda-ViT (Lin et al. (2024)) gradually degenerates MhSA blocks to identity mappings, guided by a transfer-entropy measure. Both target DeiT backbones for classification tasks.

## 1.2 ATTENTION MODELING AS CONVOLUTION

Several works have investigated the similarity between convolution and spatial relationships learned by attention. (Raghu et al. (2021)) suggests substantial differences in learned patterns, though some shallow-layer heads focus on local features. The ability of attention to capture localized patterns is further explored in Jelassi et al. (2022), emphasizing the role of positional encoding in learning spatial connectivity. Cordonnier et al. (2020) constructively proves that, under strong assumptions, an MhSA block can implement a convolutional layer if each attention head attends to a distinct location within a region the size of a convolutional kernel. In practice, as stated by the authors, this stands only as a *sufficient* condition. We further discuss this in Section 2.2. An influential

contribution to our work Han et al. (2022) discusses the relationship between attention (local) and Depth-Wise (DW) convolution, highlighting key properties that we further develop in Section 2.3.1.

### 1.3 OTHER APPROACHES

**Token Reduction.** Several works aim to reduce the number of tokens processed by transformers via removal or aggregation. DynamicViT (Rao et al. (2021)) and A-ViT (Yin et al. (2022)) dynamically select tokens to discard at each MhSA block. Token Pooling (Marin et al. (2021)) and EVIT (Liang et al. (2022)) cluster tokens into centroids, while ToMe Bolya et al. (2023) uses bipartite matching to merge token pairs. (Lu et al. (2023) )applies a policy-net to group nearby tokens by semantic class. These methods face limitations: token reduction disrupts spatial structure, making these methods unsuitable for dense prediction tasks like segmentation or depth estimation. Moreover, techniques relying on clustering, gather/scatter operations, or dynamic shapes often incur significant overhead on specialized inference frameworks.

**Efficient Attentions.** Several contributions propose alternative formulations of the attention layer to mitigate computational complexity (Shen et al. (2021); Xiong et al. (2021); Yao et al. (2024)). However, these methods are generally not intended to serve as drop-in replacements in pretrained Vision Transformers, where preserving the pretrained weights and model behavior is essential and retraining from scratch is impractical. Therefore, they are unsuitable for scenarios like ours, which require compatibility with existing pretrained models with minimal fine-tuning effort.

**Synergetic optimizations.** Alternative research directions include focusing on the algorithmic and implementation refinement of more intensive operations to take full advantage of hardware capabilities, a prominent example is Flash-Attention (Dao et al. (2022); Dao (2024)). It is also important to mention performance optimization by reduced-precision computation, combined with quantization techniques aimed at mitigating the effects of precision loss. Such techniques, with the additional use of optimized hardware (e.g, Nvidia Tensor Cores), are orthogonal to the types of approaches proposed in this work and can work in synergy to maximize speedup. We are not interested in alternative backbones such as EfficientFormer (Li et al. (2022)), the MobileViT family (Mehta & Rastegari (2022a;b); Wadekar & Chaurasia (2022)), and the recent EfficientViT (Cai et al. (2023)).

## 2 METHODOLOGY

### 2.1 BACKGROUND

**Vision Transformers and MhSA.** The ViT input is defined by splitting an image $I \in \mathbb{R}^{H \times W \times 3}$ into non-overlapping patches of size $p \times p$, flattening each patch into a vector in $\mathbb{R}^{3p^2}$ and projecting it to a $d$-dimensional embedding. Assuming $\frac{H}{p} = \frac{W}{p} = \sqrt{n}$, this yields an input tensor $X \in \mathbb{R}^{n \times d}$ of $n$ patch embeddings, to which is added a positional encoding. Most implementations, including Dino-V2, prepend a `[cls]` token, which we ignore for now. The sequence is processed through transformer blocks alternating Multi-head Self-Attention (MhSA) and feed-forward layers.

Let $n_h$ be the number of attention heads, the MhSA is parametrized by $W^Q, W^K, W^V \in \mathbb{R}^{d_i \times (n_h * d_h)}$ and $W^O \in \mathbb{R}^{(d_h * n_h) \times d_o}$ where typically $d_i = d_o = d$ and $d_h * n_h = d$, as we assume from now on. Defining Query, Keys, and Values for the $h$-th head as:

$$Q^h = XW^Q_{[:,h,:]} K^h = XW^K_{[:,h,:]} V^h = XW^V_{[:,h,:]} \tag{1}$$

For the $h$-th head, the attention is computed as:

$$Att(X)^h = E(X)^h V^h \tag{2}$$

$$E(X)^h = softmax\left(\frac{Q^h K^{h\top}}{\sqrt{d_h}}\right) \tag{3}$$

where $E(X)^h \in \mathbb{R}^{n \times n}$ is the attention (energy) matrix. Concatenating the head outputs yields the full MhSA output:

$$MhSA(X) = [Att^1(X) \| \dots \| Att^{n_h}(X)]W^O \tag{4}$$

with $\|$ denoting concatenation along the embedding dimension.

**Convolutional Layers.** A Convolutional layer with kernel size $k$ is parametrized by weights $W^C \in R^{k \times k \times c_i \times c_o}$. Expliciting the symmetric shift set of size $k \times k$ as $\Delta_k = \{(s, r) \in \mathbb{Z}^2 : -\lfloor k/2 \rfloor \leq s, r \leq \lfloor k/2 \rfloor\}$ (assuming a stride factor of 1), the output at location $(i, j)$ for an input $X \in R^{h \times w \times c_i}$ is defined as:

$$Conv(X, W^C)_{i,j} = \sum_{r,s \in \Delta_k} W_{r,s}^{C^\top} X_{i+r,j+s} \tag{5}$$

Thus, convolution produces a weighted local aggregation of the $k \times k$ neighborhood centered at $(i, j)$. Similarly, depth-wise convolution performs local aggregation by applying a distinct spatial filter to each channel independently. With $W^D \in \mathbb{R}^{k \times k \times c_i}$, we write

$$Conv_{DW}(X, W^D)_{i,j} = \sum_{(r,s) \in \Delta_k} W_{r,s}^D \odot X_{i+r,j+s} \tag{6}$$

where $\odot$ denotes elementwise multiplication.

## 2.2 CONVOLUTIONAL APPROXIMATION

In this section, we formalize how attention can be approximated by convolution, and present the efficient depthwise decomposition at the core of our method. We later show that the SPViT He et al. (2024) bottleneck block arises as a special case of our formulation, extended with head ensembling.

### 2.2.1 DROP-IN DEPTHWISE FORMULATION.

Consider a single attention head as in Equation (2). For clarity, we reshape the input $X \in \mathbb{R}^{n \times d}$ to $X \in \mathbb{R}^{m \times m \times d}$ with $m = \sqrt{n}$, recovering spatial structure. Accordingly, $E(X)^h \in \mathbb{R}^{n \times n}$ can be viewed as $E(X)^h \in \mathbb{R}^{(m \times m) \times (m \times m)}$, and the values as $V^h \in \mathbb{R}^{m \times m \times d_h}$. When writing two-dimensional indices, we always refer to these unflattened tensors. The explicit form of attention at location $(i, j)$ is:

$$Att(X)_{i,j}^h = \sum_{r,s \in \Delta_m} E^h(X)_{(i,j),(i+r,j+s)} V_{i+r,j+s}^h \tag{7}$$

where $\Delta_m$ denotes the full receptive field that spans the whole $E(X)^h$. This resembles convolution, with key differences: the spatial aggregation weights $E^h(X)_{(i,j),(i+r,j+s)}$ depend on both the input $X$ and the query position $(i, j)$, and convolutional kernels are fixed parameters shared across locations.

We approximate attention by assuming that some heads can be replaced by *input-independent kernels* restricted to a local neighborhood $\Delta_k \subset \Delta_m$. Formally, for head $h$ we write:

$$Att(X)_{i,j}^h \approx \sum_{(r,s) \in \Delta_k} K_{r,s}^h V_{i+r,j+s}^h \tag{8}$$

where $K^h \in \mathbb{R}^{k \times k}$ are trainable parameters learned during fine-tuning.

**Full convolution formulation.** A direct implementation of Equation (8) is to fold $K^h$ into the value projection $W^V$, producing a kernel $W^{Vh} \in \mathbb{R}^{k \times k \times d_i \times d_h}$:

$$\widetilde{Att}_C(X)^h = Conv(X, W^{Vh}) \tag{9}$$

$$W_{r,s}^{Vh} = K_{r,s}^h W_{[:,h,:]}^V, \quad (r, s) \in \Delta_k \tag{10}$$

This formulation, while being a faithful analogue of Eq. (8), is not appealing from a complexity standpoint, as assessed in Section 3.2.

**Depthwise decomposition.** To reduce cost, we separate the pointwise value projection from the spatial aggregation. We first compute values $V^h = X W_{[:,h,:]}^V$, then apply a depthwise convolution with head-specific kernels $\vec{K}^h \in \mathbb{R}^{k \times k \times 1 \times d_h}$:

$$\boxed{\widetilde{Att}_{DW}(X)^h = Conv_{DW}(V^h, \vec{K}^h)} \tag{11}$$

Compared to the full convolution, the complexity is reduced from $O(k^2 d_i d_h)$ to $O(d_i d_h + k^2 d_h)$. In the full formulation (Equation (9)), each $K_{r,s}^h$ is shared across all $d_h$ channels, enforcing a single spatial pattern. Depthwise kernels $\vec{K}^h$, instead, provide one $k \times k$ filter per channel, enabling distinct spatial aggregations. While hannel-wise sharing could be easily implemented, we relax it without affecting performance. This layer can replace any subset of attention heads in the MhSA block (Eq. (4)). Its implementation is schematized in Figure 1 and thoroughly evaluated in the experimental section.

**Head-ensembling.** As detailed in Section A.1.1, the formulation of He et al. (2024) builds on the sufficient condition of Cordonnier et al. (2020), where each head attends to a distinct spatial location within a local neighborhood. This assumption implicitly enforces a degenerate head ensembling, causing the MhSA block to collapse into a single effective head. In contrast, we derive the ensembling explicitly and show that it extends beyond the full convolution setting. Specifically, by assigning learnable combination weights $\gamma \in \mathbb{R}^{n_h}$ to control the contribution of each head, the ensembled value and output projections become:

$$W^{Ve} = \sum_{h=1}^{n_h} \sigma(\gamma_h) \, W_{[:,h,:]}^V, \qquad W^{Oe} = \sum_{h=1}^{n_h} \sigma(\gamma_h) \, W_{[:,h,:]}^O, \tag{12}$$

with $\sigma(\cdot)$ a softmax over heads. Crucially, this explicit ensembling extends naturally to our depthwise formulation:

$$\widetilde{MhSA}_{DW}^e(X) = Conv_{DW}(V^e, \vec{K}^e)W^{Oe} \tag{13}$$
$$s.t \quad V^e = XW^{Ve}$$

and $\vec{K}^e$ denotes the depthwise convolution kernel as in Equation (11). In this view, He et al. (2024) arises as a *special case* of our framework. We henceforth distinguish between the ensembled formulation (Equation (13)) and the unensembled formulation (direct head replacement via Equation (11)).

## 2.3 LAYER SELECTION

Given a target of $p_h$ heads to approximate with convolution, selection can be either scattered, replacing arbitrary heads across the model, or blockwise, replacing all $n_h$ heads within $p_b$ MhSA blocks. As shown in Sec. 3.3, the blockwise strategy yields higher inference efficiency, and we therefore adopt it as the default. Below, we introduce two criteria for defining the head set $\mathcal{S}$, both applicable to either selection mode.

### 2.3.1 PROPOSED CRITERION.

As briefly mentioned, the approximation for $E(X)^h$ introduced in Equation (8) is equivalent to real attention under the conditions of Locality (L), translation invariance (TI), and Input Invariance (II). For a receptive field $\Delta_k$ and displacement $(s, r) \in \Delta_k$, these are:

$$\begin{cases} (\text{L}) : E(X)_{(i,j),(u,v)}^h \neq 0 \;\; \text{only if } (u-i, v-j) \in \Delta_k & (14a) \\[6pt] (\text{TI}) : E(X)_{(i,j),(i+s,j+r)}^h = E(X)_{(l,t),(l+s,t+r)}^h \quad \forall (i,j),(l,t) & (14b) \\[6pt] (\text{II}) : E(X)^h = E(Y)^h \quad \forall X, Y & (14c) \end{cases}$$

**Criterion Definition** We empirically establish the sum of the pointwise standard deviation of $E(X)^h$ as a simple and effective proxy for identifying convolutional-like heads. Concretely, for each head $h \in \{1, \ldots, N_h\}$, where $N_h = n_h n_b$ is the cumulative number of heads across $n_b$ blocks, we compute the pointwise standard deviation $\sigma_{E^h}$ of $E(X)^h$ over $N_s$ input samples. Direct computation of $\sigma_{E^h}$ impractical, since accumulating $E^h(X_i)$ for a reasonable input set (i.e., $N_s = 1000$) would require over 600GB of memory. We use Welford's algorithm Welford (1962) to compute $\sigma_{E^h}$ online in a single pass. We then define a scalar score

$$\Sigma_h = \sum \sigma_{E^h}, \tag{15}$$

summing over all entries of $E^h$. We select as candidate heads the set $\mathcal{S}_h^{[p_h]}$ of the $p_h$ heads with the smallest $\Sigma_h$. In the blockwise setting, we adopt the same criterion at block level. For each block $b \in \{1, \ldots, n_b\}$ we compute the mean score across its $n_h$ heads:

$$\Sigma_b = \frac{1}{n_h} \sum_{h \in [n_h]} \Sigma_h, \tag{16}$$

and select the set $\mathcal{S}_b^{[p_b]}$ of $p_b$ blocks with the smallest $\Sigma_b$.

**Rationale**  By construction, $\Sigma_h = 0$ is both necessary and sufficient for the Input-Invariance property Equation (14c). In the limit $\Sigma_h \to 0$, the kernel collapses to its expectation $E^h(X) \to \mu_{E^h}$, eliminating dependence on the input. Our heuristic is driven by the observation that Input-Invariance is the most stringent property: once achieved, it forces the head to ignore input variation entirely and reduces to a positional-only operator. In this regime, $E^h(X)$ derives solely from the learned positional encodings shared across heads in standard ViTs. Positional attention mechanisms have been linked to spatial connectivity patterns (patch association; Jelassi et al. (2022)), capturing the locality and translation-like structure that underlies convolutional inductive biases. We therefore use $\Sigma_h$ as a heuristic, motivated by the Input-Invariance principle, but ultimately empirical. In practice, we find it to be a simple and effective selection rule, with extensive experimental validation in the remainder of the paper and additional visualizations in the Appendix (Figure A.2).

### 2.3.2 STOCHASTIC GATING.

As an alternative to the presented criterion, we propose a comparison with a selection method derived from Differentiable Subset Pruning (DSP) Li et al. (2021). While DSP in origin prunes transformer heads, we can easily generalize it for our scope. For simplicity, we present this mechanism in the blockwise case, although it can be trivially extended to the scattered selection. We define a set of trainable parameters $w^b \in \mathbb{R}^{n_b}$, leveraging the topk$(\cdot)$ operator we can retrieve the largest $p_b$ elements of $w^b$:

$$\text{topk}(w^b, p_b)_i = \begin{cases} 1 & \text{if} \quad i \in \mathcal{S}_b^{[p_b]} \\ 0 & \text{otherwise} \end{cases} \tag{17}$$

Defining $\overline{w}_i^b = \text{topk}(w^b, p_b)_i$, we can implement a simple gating mechanism to select the chosen operation during the forward pass:

$$\overline{MhSA}^i(X) = (1 - \overline{w}_i^b)MhSA(X)^i + \overline{w}_i^b M\widetilde{h}SA(X)^i \tag{18}$$

This formulation involves only a minimal increase in the number of parameters and no additional loss terms. Since the topk operator is non-differentiable, to learn $w^b$ during training, the Gumbel top-k relaxation Kool et al. (2019) is used, an extension of the Gumbel softmax trick Jang et al. (2016), which provides a differentiable approximation $\tilde{w}^b$ of the hard selection $\overline{w}^b$, controlled by a temperature $\tau$. As $\tau \to 0$, $\tilde{w}^b$ approaches $\overline{w}^b$. We begin training with a higher $\tau$ to enable gradient flow, then anneal it to $10^{-3}$ using the schedule from DSP (Section 3.3).

## 3 EXPERIMENTS AND COMPARISONS

### 3.1 EXPERIMENTAL SETUP

Unless otherwise specified, our analysis is based on the Dinov2 (Oquab et al. (2023)) model fine-tuned on various downstream tasks. This choice simplifies the discussion of the results. In Section 3.2, we show that similar results hold when using other vision foundation transformer backbones.

**Benchmarking.**  To reflect real deployment conditions, we mainly target TensorRT on Nvidia Jetson Orin Nano to profile inference performances. Unlike general-purpose frameworks (e.g., PyTorch), TensorRT compiles models into optimized GPU kernels, highlighting real-world performance constraints. In the appendix, we further detail the profile setup (Section A.2.2) and later extend the analysis to a broader range of hardware and software architectures (Section A.4.1).

**Task Performance.** We evaluate fine-tuned models on semantic segmentation (COCO Lin et al. (2014), ADE20K Zhou et al. (2017)) and classification (ImageNet-1K Russakovsky et al. (2015)). To support drop-in convolutions, we remove the [cls] token from inputs. For segmentation, this has no impact; for classification, we use the mean of tokens as decoder input, with negligible performance loss. Convolutional layers use a fixed kernel size $k = 3$ for efficiency across all tasks.

## 3.2 CONVOLUTION ATTENTION RESULTS

### 3.2.1 HEAD-LEVEL PROFILING.

Table 1: Comparison of different choices for the MhSA block. All results are measured for a single MhSA block with $n_h = 16$ heads and an input size of $24 \times 24$ patches (Equivalent to an image resolution of $336 \times 336$ for Dino-V2). Batch size is set as 1.

| Attention | FLOPs (G) | Params (M) | Inference (ms) | Memory (MB) |
|---|---|---|---|---|
| MhSA (Eq. (4)) | 6.19 | 4.2 | 3.2 | 47.2 |
| Conv$_{[all]}$ (Eq. (9)) | 12.08 | 2.1 | 3.71 | 4.5 |
| DW$_{[all]}$ (Eq. (11)) | 2.43 | 2.11 | **1.26** | 6.75 |
| SPViT-*style* (He et al. (2024)) | 0.75 | 2.1 | 0.641 | 4.5 |
| Ens+DW (Eq. (13)) | 0.15 | 2.1 | **0.215** | 0.288 |

In Table 1, we profile a single multi-head attention block, providing a straightforward setting to quantify performance differences. Both in the unensembled setup (lines 2-3) or in the ensembled one (lines 4-5), it is clear that the depthwise formulation is advantageous, speeding up execution by a factor $3\times$ with respect to the full convolution. The ensembled formulation is significantly faster, although, as discussed later in this chapter, it results in a more significant performance drop. Memory usage patterns are less intuitive; the separable formulation has a slightly higher memory requirement due to intermediate results and less effective buffer reuse, which is offset in the ensembled formulation.

### 3.2.2 FULL MODEL PERFORMANCE.

When not otherwise specified, we first perform fine-tuning on the target task with regular MhSA heads, replace the selected heads with convolutional layers, and perform a new fine-tuning for half of the training epochs. The complete experimental setup is detailed in the Appendix. In Table 2 we compare the results obtained with different options for the MhSA block. For all experiments, we use the standard deviation criterion proposed in Section 2.3 with the blockwise selection. Despite not aiming for state-of-the-art performance, the strength of Dinov2 features makes our baseline results highly competitive. Latency is reported only for the ViT backbone to avoid being affected by the design choices of the decoder.

**Evaluation** We first confirm that the depthwise formulation matches the performance of full convolution (*CL2–CL3*), in line with our analytical derivation, while delivering a substantial speedup in inference. Without head ensembling, the full convolution baseline is 7.5% slower than MhSA (*CL1*), whereas the depthwise variant achieves a 17.2% speedup. For a fair comparison with the ensembled setup, we match configurations of $|\mathcal{S}|$ with similar observed speedup $|\mathcal{S}|$: $12/24$ blocks for the unensembled case and $10/24$ for the ensembled case. In this setting, depthwise unensembled (*CL3*) incurs a smaller accuracy drop (-0.08 vs. -0.98 mIoU) than the SPViT-style full-convolution ensemble, while still providing a modest speedup advantage. When combined with head ensembling (*CL5*), the depthwise formulation achieves an 18.8% speedup, but at the cost of a larger -1.6 mIoU drop. The same trade-off is observed when increasing $|\mathcal{S}|$ to $16/24$ and $12/24$ (*CL6–CL7*). Comparable results are observed on the smaller ViT-B backbone (*CB1–CB3*) and on the ImageNet classification task (*IL1–IL5*), confirming the consistency of these observations across model scale and heterogeneous tasks. Additional results on ViT-B and on ADE20K dataset are reported in Table 3

Table 2: Results on COCO and Imagenet with different formulations. Results are obtained finetuning Dino-V2, $336 \times 336$ input resolution. Inference performances are reported with batch-size=1.

| ID | Task | ViT | Attention | $|\mathcal{S}|$ | mIoU | $\delta$-mIoU | Infer (ms) | Speedup (%) |
|---|---|---|---|---|---|---|---|---|
| CL1 | | | MhSA | - | 66.03 | (baseline) | 161.4 | (baseline) |
| CL2 | | | Conv$_{[all]}$ | 12/24 | 65.84 | -0.19 | 173.5 | -7.49 |
| CL3 | | Large | DW$_{[all]}$ | 12/24 | 65.95 | -0.08 | 133.6 | 17.21 |
| CL4 | | | SPViT-*style* | 10/24 | 65.05 | -0.98 | 133.3 | 17.40 |
| CL5 | COCO | | Ens + DW | 10/24 | 64.81 | -1.61 | 131.1 | 18.79 |
| CL6 | | | DW$_{[all]}$ | 16/24 | 64.64 | -1.39 | 126.1 | 21.85 |
| CL7 | | | Ens + DW | 12/24 | 63.92 | -2.11 | 124.5 | 22.87 |
| CB1 | | | MhSA | - | 63.37 | (baseline) | 49.9 | (baseline) |
| CB2 | | Base | DW$_{[all]}$ | 6/12 | 62.22 | -1.16 | 40.8 | 18.10 |
| CB3 | | | Ens + DW | 6/12 | 60.55 | -2.83 | 38.1 | 23.52 |

| ID | Task | ViT | Attention | $|\mathcal{S}|$ | Top-1 Acc. | $\delta$-Acc | Infer (ms) | Speedup (%) |
|---|---|---|---|---|---|---|---|---|
| IL1 | | | MhSA | - | 86.22 | (baseline) | 161.4 | (baseline) |
| IL2 | | | DW$_{[all]}$ | 12/24 | 85.45 | -0.77 | 133.6 | 17.21 |
| IL3 | Imagenet | Large | Ens + DW | 10/24 | 84.96 | -1.26 | 131.1 | 18.79 |
| IL4 | | | DW$_{[all]}$ | 16/24 | 84.88 | -1.34 | 126.1 | 21.85 |
| IL5 | | | Ens + DW | 12/24 | 84.65 | -1.57 | 124.5 | 22.87 |

**Generalization.** We evaluate the generalization of our approach to backbone models beyond Dinov2 by considering CLIP (Radford et al. (2021)) and MAE (He et al. (2022)). Despite their different pretraining objectives, both models share the same underlying ViT architecture, allowing us to replicate our training setup without modification. For each model, we evaluate both the ViT-Base and ViT-Large variants, with patch sizes indicated by /16 and /14, respectively. Results are reported in Table 4. We use the unensembled formulation with blockwise selection, applying the $\Sigma_b$ criterion to identify the block set. Observing the drop in mIoU relative to the same backbone without drop-in convolutions (i.e., $|\mathcal{S}| = -$), we observe results that mirror those obtained with Dinov2, further validating the generality of our drop-in formulation across different vision foundation models.

Table 3: Results on different tasks using the Depthwise formulation.

| Task | ViT | Attention | $|\mathcal{S}|$ | Acc. |
|---|---|---|---|---|
| **Imagenet** | **Base** | MhSA | - | 83.76 |
| | | DW$_{[all]}$ | 6/12 | 82.00 |

| Task | ViT | Attention | $|\mathcal{S}|$ | mIoU |
|---|---|---|---|---|
| **ADE20K** | **Base** | MhSA | - | 53.83 |
| | | DW$_{[all]}$ | 6/12 | 51.70 |
| | **Large** | MhSA | - | 56.84 |
| | | DW$_{[all]}$ | 12/24 | 56.05 |

Table 4: Evaluation of Depthwise formulation applied to MAE and CLIP fine-tuned on COCO Semantic segmentation.

| Model | Backbone | $|\mathcal{S}|$ | mIoU | Infer (ms) |
|---|---|---|---|---|
| MAE | ViT-B/16 | - | 58.84 | 42.95 |
| MAE | ViT-B/16 | 6/12 | 57.91 | 35.04 |
| MAE | ViT-L/16 | - | 60.22 | 122.47 |
| MAE | ViT-L/16 | 12/24 | 59.81 | 102.23 |
| CLIP | ViT-B/16 | - | 61.52 | 42.93 |
| CLIP | ViT-B/16 | 6/12 | 59.06 | 35.07 |
| CLIP | ViT-L/14 | - | 64.65 | 153.91 |
| CLIP | ViT-L/14 | 12/24 | 62.17 | 128.38 |

## 3.3 SELECTION MECHANISM RESULTS

We first discuss the implications of blockwise and scattered selection. While for blockwise selection the exact subset $\mathcal{S}_b^{[p_b]}$ has no impact on the observed speedup, in the scattered setting the distribution of $\mathcal{S}_h^{[p_h]}$ is relevant. Scattered heads selection causes speedup to scale non-linearly with the number of selected heads due to overhead from memory and multiple kernels execution, which can offset gains. The impact on model performance is clearly observable in Figure 2, with the blockwise selection being consistently faster, while only implying a small performance drop (Table 5).

**Comparison.** In Table 5 we compare the proposed selection heuristic (Section 2.3.1), with differentiable subset pruning (DSP) Section 2.3.2. For the latter, we evaluate both end-to-end training of selection gates (DSP-e2e, Equation (18)) and a two-stage variation (DSP-2S), where the learned set

Table 5: Comparison of selection mechanisms.

| Crit. | Stages | BW | $|\mathcal{S}|$ | mIoU |
|---|---|---|---|---|
| $\Sigma_{b[\texttt{LOWEST}]}$ | 2 | | 12/24 | 65.95 |
| DSP - e2e | 2 | Yes | 12/24 | 64.15 |
| DSP - 2S | **3** | | 12/24 | 64.00 |
| $\Sigma_{b[\texttt{LOWEST}]}$ | 2 | | 17/24 | 63.77 |
| DSP - e2e | 2 | Yes | 17/24 | 58.25 |
| DSP - 2S | **3** | | 17/24 | 64.20 |

| Crit. | Stages | BW | $|\mathcal{S}|$ | mIoU |
|---|---|---|---|---|
| $\Sigma_{h[\texttt{LOWEST}]}$ | 2 | | 192/384 | 65.52 |
| DSP - e2e | 2 | No | 192/384 | 62.32 |
| DSP - 2S | **3** | | 192/384 | 65.79 |
| $\Sigma_{b[\textbf{HIGHEST}]}$ | 2 | | 12/24 | 61.46 |
| $\Sigma_{b[\textbf{HIGHEST}]}$ | 2 | Yes | 7/24 | 64.02 |
| $\Sigma_{b[\texttt{LOWEST}]}$ | **1** | | 12/24 | 65.63 |

(a) Comparision between $\Sigma_b$ and DSP in Blockwise setting

(b) (top) Comparision between $\Sigma_h$ and DSP in Scattered setting (bottom) Ablation of $\Sigma_b$: selection of the worst candidates ($\Sigma_{b[\textbf{HIGHEST}]}$) and 1-stage finetuning.

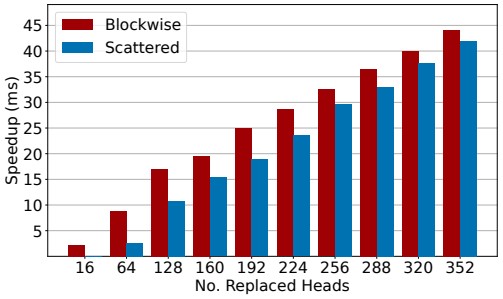

Figure 2: Speedup vs number of heads replaced in blockwise and scattered setups. Results on ViT-L (24 blocks, 16 heads per block $336\times336$).

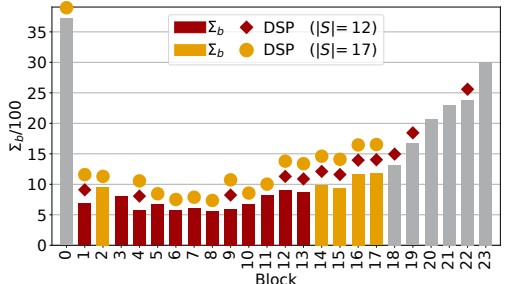

Figure 3: Blocks selected using DSP and $\Sigma_b$ criterion in blockwise setting.

is reused in a new fine-tuning run with fixed selection, discarding the end-to-end weights. In the blockwise setup (Table 5a) with $|\mathcal{S}| = 12/24$ the $\Sigma_b$ criterion outperforms the DSP, and DSP-2S reduces the gap only slightly, still trailing by nearly 2 mIoU points. A closer inspection (Figure 3), reveals that DSP often selects high-variance heads. Increasing to $|\mathcal{S}| = 17/24$ DSP-2S slightly surpasses $\Sigma_b$, while DSP-e2e suffers from severe training instabilities and degraded performance. In Table 5b (top section), this analysis is extended to the scattered selection, with a similar outcome which is further assessed in Section A.3.3.

**Ablation.** To further test our criterion, we also evaluate replacing the worst heads (highest $\Sigma_{E^h}$) in Table 5b. With only 7 blocks replaced, the performance drop already exceeds that of the best 12 blocks, and replacing the 12 worst blocks leads to severe degradation. Finally, we ablate the fine-tuning procedure (last row): a single-stage fine-tuning, where convolutions are applied directly, achieves performance close to the two-stage setup. Further ablations, such as fine-tuning only the convolutional kernel weights, are discussed in Section A.3.

## 4 CONCLUSIONS AND FUTURE WORK

We assessed the effectiveness of a simple drop-in replacement for attention heads exhibiting convolution-like behavior in large-scale pretrained ViTs. The proposed framework achieved 17% speedup with minimal impact on downstream performance, highlighting that many pretrained heads can be approximated by efficient depthwise convolution without losing their functional role and thus largely preserving the power of pretrained weights. The investigated approach is not an alternative to existing pruning techniques, but rather as a component that can also be effective in combination with existing solutions. We will investigate this direction in future work. The selection criterion is another direction that requires further investigation. The proposed heuristic proved to be very effective, given its simplicity, yet there is clearly room to investigate more advanced selection criteria.

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

## A.1 ADDITIONAL DETAILS

### A.1.1 DERIVATION OF ENSEMBLED FORMULATION

A closely related formulation is discussed in He et al. (2024), to derive it, we can rewrite Equation (4) as:

$$MhSA(X) = \sum_{h \in [n_h]} Att^h(X) W_{[:,h,:]}^O \tag{A.1}$$

If we apply the approximation $\widetilde{Att}$ from Equation (8) to all attention heads, we can rewrite explicitly the approximated $MhSA$ output value at spatial location $(i, j)$ as:

$$\widetilde{MhSA}(X)_{i,j} = \sum_{h \in [n_h]} \left( \sum_{(r,s) \in \Delta_k} K_{r,s}^h X_{\substack{i+r, \\ j+s}} \right) W_{[:,h,:]}^V W_{[:,h,:]}^O \tag{A.2}$$

The authors construct their formulation based on Cordonnier et al. (2020), which derives a *sufficient* condition valid only in the case where each attention head attends at one and only one spatial location in $\Delta_k$, under the assumption that $k = n_h$. For ease of notation, we can imply this condition by imposing:

$$K_{(s,r)}^h \neq 0 \leftrightarrow sk + r = h$$
$$k = n_h \tag{A.3}$$

Only under a similar assumption, Equation (A.2) can be equivalent to:

$$\widetilde{MhSA}_{i,j}^e = \left( \sum_{(s,r) \in \Delta_k} X_{i+s,j+r} \overline{W^V}_{s,r} \right) \overline{W^O} \tag{A.4}$$

$$s.t \quad \begin{cases} \overline{W^V}_{s,r} = \sum_{h \in [n_h]} K_{s,r}^h W_{[:,h,:]}^V \\ \overline{W^O} = \sum_{h \in [n_h]} \sum_{(s,r) \in \Delta_k} K_{s,r}^h W_{[:,h,:]}^O \end{cases}$$

An assumption equivalent to Equation (A.3), imply $\overline{W^O} = W^O$ and $X_{i+s,j+r} \overline{W^V}_{s,r} = X_{i+s,j+r} W_{[:,h,:]}^V$, $(h = sk + r)$. In practice, this collapses the MhSA block to a single convolution, therefore, the same implementation of Equation (11) can be applied to cast the innermost product of Eq. (A.4). The authors of He et al. (2024) implicitly relax both conditions of Eq. (A.3) and learn to ensemble the attention heads controlled by the weights $\sigma_h(K)$ where $\sigma_h(\cdot)$ defines the softmax function applied over the heads dimension. We defer to the referring work for further details. In Equation (A.4) the parameters $K \in \mathbb{R}^{n_h \times k \times k}$ control both the ensembling of the multiple heads in a single one and the spatial aggregation through the convolution operation. By defining a set of learnable parameters $\gamma \in \mathbb{R}^{n_h}$ to control the aggregation of heads projections, the formulation of Equation (A.4) can be seen as a special case of our method presented in Section 2.2.1, with the addition of an ensembling of the head projections controlled by a set of trainable parameters.

## A.2 EXPERIMENTAL SETUP

### A.2.1 DOWNSTREAM FINETUNING SETUP

For all downstream tasks, we share a similar optimization setup, leveraging AdamW optimizer with a Cosine annealing scheduler and linear warmup. For each task, we build a lightweight decoder on top of pretrained Dino-V2, finetuning Dino while training the decoder from scratch. We assess performance using established metrics from the literature: mean Intersection over Union (mIoU) for semantic segmentation and top-1 accuracy, based on the highest-scoring prediction, for image classification.

**Semantic segmentation.** The semantic segmentation task is a good representative of dense prediction tasks, and the complexities of formulations found in tasks like instance segmentation. We build a simple convolutional decoder on top of Dino-v2 backbone to upscale the patch-level embeddings

from DINO to pixel-level embeddings. The decoder consists of 3 transposed convolution blocks, with Group Normalization Wu & He (2018) and GeLU activation Hendrycks & Gimpel (2016), to gradually recover the input resolution, and two additional convolution layers project pixel-level embeddings to the required number of channels. For all experiments on COCO, we use an input resolution of $336 \times 336$ and $192 \times 192$ output resolution. For ADE20K, the input and output resolutions are instead $336 \times 448$ and $192 \times 256$. We optimize the CrossEntropy loss function, first stage finetuning is conducted for 100 epochs with an effective batch-size of 1024, leveraging random resized crop and color jitter augmentation.

**Image Classification.** We follow the same design philosophy for the classification task on Imagenet-1K. As done in the DinoV2 paper, we build a minimal decoder with a single fully-connected layer to project $d$ embedding dimension to 1000 classes logits. When cls token is not available, we use the average of all output tokens as a substitute. We optimize cross-entropy loss and use only random crop and horizontal flip as training-time augmentation. First-stage finetuning is conducted for 50 epochs.

**Finetuning with Convolution.** After the first finetuning stage, we replace the selected heads with the formulation of choice, query and key projection weights are discarded. The second finetuning stage is performed for half of the first stage epochs with half batch size and the same hyperparameters. For convolutional layers, we initialize kernel weights with a Gaussian distribution, having observed a slight speedup in convergence speed, and exclude kernel parameters from L2-regularization.

### A.2.2 BENCHMARKING SETUP

**Nvidia Hardware.** To profile models on Nvidia devices, we use the classic workflow of exporting from pytorch to onnx format and then building the TensorRT engine leveraging Python APIs, serializing the produced engine to a file. For benchmarking, we leverage the provided `trtexec` utility, with the following set of arguments:

```
$ trtexec --loadEngine=model.onnx --useCudaGraph --noDataTransfers
--useSpinWait --iterations=100 --avgRuns=100 --exportTimes=measure.json
```

For detailed head profiling (Table 1, Table A.4) we leverage the Nnvidia DL Designer tool[1], providing the onnx model.

**Additional hardware.** To profile on HAILO8 we first compile onnx models to `.hef` (Hailo Executable Format) binaries leveraging the provided sdk, then run benchmarking on the target hardware with the included profiling utility. For CPU platforms, we directly run the onnx model with onnx Runtime, leveraging a custom python script. To avoid fluctuations due to the operating system scheduler, we pin the profiling process to a single core and execute with a real-time scheduling policy. This is achieved with the following syntax:

```
$ taskset -c 3 chrt -f 99 python3 benchmark_onnx.py <model_file>.onnx
```

## A.3 ADDITIONAL EXPERIMENTS

### A.3.1 FINETUNING CONVOLUTION ONLY

We briefly experimented with performing second-stage finetuning by freezing first-stage weights, except for the affected heads. The preliminary experiment, reported in Table A.1 performs surprisingly well, considering only $1/10$ of the parameters are updated.

### A.3.2 DEALING WITH [CLS] TOKEN

As mentioned in the discussion, to make Dino-V2 ViT backbone compatible with the convolutional formulation, we need to remove the [cls] token, which would not allow reshaping value tokens. This could be a concern,

---

[1]https://developer.nvidia.com/nsight-dl-designer

Table A.1: Results on COCO for second-stage finetuning (ViT-L, Conv-DW) Frozen or Unfrozen Backbone

| Task | Frozen BB | Train Params | Metric |
|------|-----------|--------------|--------|
| COCO | NO | 284.4M | 65.95 |
| COCO | YES | 25.3M | 65.12 |

since [cls] is used in pretraining and typically in classification tasks. In Table A.2 we show a quick ablation on COCO and Imagenet tasks, for the former we observe no impact on mIoU metric, the latter has a small drop in Accuracy. In light of these results, we opted for the most direct approach, removing the [cls] from the model.

Table A.2: Effect of `[cls]` token presence on different tasks

| Task | ViT | Attention | [CLS] | Metric |
|------|-----|-----------|-------|--------|
| COCO | Large | MhSA | NO | 66.03 |
| COCO | Large | MhSA | YES | 66.03 |
| Imagenet | Large | MhSA | NO | 86.22 |
| Imagenet | Large | MhSA | YES | 86.35 |

In a preliminary stage, we considered the option of leveraging a set of parameters to control the flow of global information to the [cls] token, we did not pursue this direction because of the overhead.

### A.3.3 CONSIDERATIONS ON SELECTION CRITERIA

For the proposed criterion, we compute $\Sigma_h$ (Equation (15)) and consequently $\Sigma_b$ after the first stage finetuning, leverage $N_s = 1000$ samples from the training split of the corresponding dataset.

**Effects of Finetuning.** We briefly discussed the possibility of single-stage finetuning, applying convolution over Dino weights, with no prior finetuning on the target task. For the experiment reported in Table 2, we observe promising results, while still relying on finetuned weights to apply $\Sigma$ criterion. In Figure A.1, we report $\Sigma_b$ distribution before and after finetuning on COCO, we can observe that selection using $\Sigma$ criterion is not significantly affected by finetuning. With $n_b = 17$ the selection set would match, with $n_b = 12$ the selection differs by a single block. This suggests further potential to investigate single-stage finetuning.

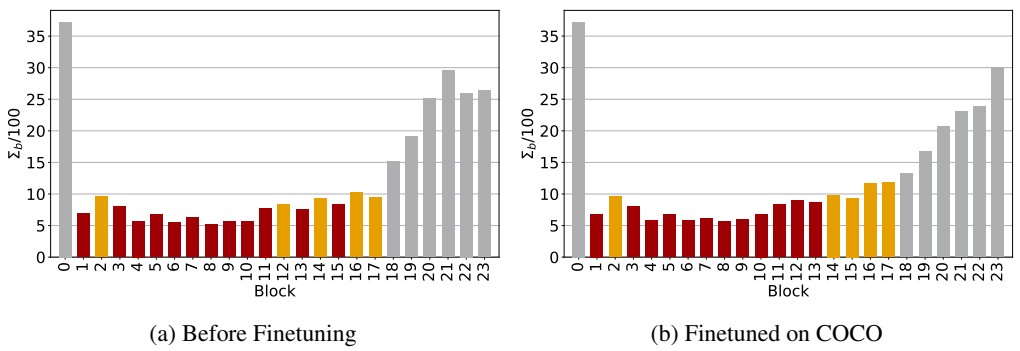

(a) Before Finetuning

(b) Finetuned on COCO

Figure A.1: Distribution of $\Sigma_b$ before and after COCO finetuning. In red, the 12 selected blocks, in orange, the next 5 (selecting 17 blocks)

**Visualizations.** In Section 2.3.1, we introduced the criterion based on standard deviation for selection of heads to be replaced. In Figure A.2 we propose an intuitive visualization to qualitatively appreciate the effect of $\sigma_h$ on the attention kernels. We obtain the visualization by aligning each element of reshaped $\sigma_h \in \mathbb{R}^{(m \times m) \times (m \times m)}$ (i.e., each row of $E(X)^h$) by centering the query pixel at a fixed location (center pixel), and finally computing the mean value. This visualization allows us to highlight that the relationship between sigma and locality properties is independent of input.

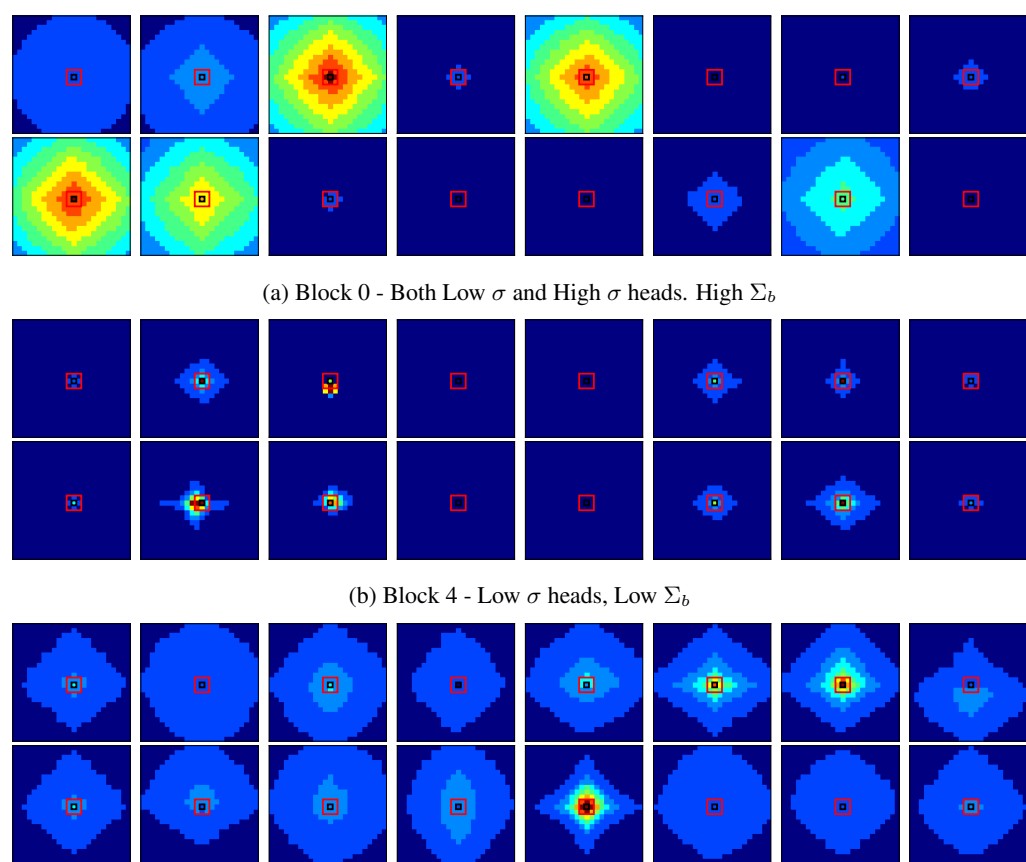

(a) Block 0 - Both Low $\sigma$ and High $\sigma$ heads. High $\Sigma_b$

(b) Block 4 - Low $\sigma$ heads, Low $\Sigma_b$

(c) Block 22 - High $\sigma$ heads, High $\Sigma_b$

Figure A.2: Visualization of $\sigma_h$ for different heads in Dino-V2 ViT-L finetund on coco datasets. The visualization is obtained by aligning all $\sigma_h$ around the central pixel and computing the average. Red square represents the size of the $3 \times 3$ convolutional kernel.

**Contribution of Positional Encoding.** In Section 2.3 we hinted that a key role is played by positional encoding in enforcing convolutional-like behavior when $\Sigma_h \to 0$. In an attempt to get an insight, in Section A.3.3 we visualize the correlation score between positional encoding vectors.

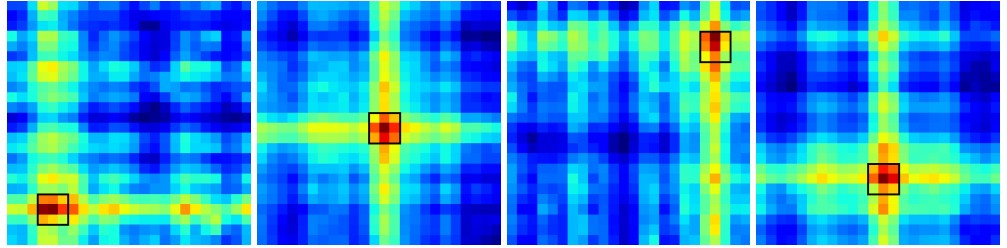

Figure A.3: Visualization of correlation between Dino-V2 positional embedding vectors at sample query locations.

This simple experiment provides insight into the strong spatial patterning and locality bias induced by positional encoding.

**Distribution of selected Heads.** In Section 3.3 we compared $\Sigma$ and DSP criteria and discussed that in the blockwise setup, DSP tends to agree with $\Sigma$ when performing competitively on the task ($n_b = 17$), as shown Figure 3. We extend this visualization to the scattered case in Figure A.4, in this case we note that although the two methods perform similarly (Table 5), the selected heads follow a different distribution. So far, we can only

speculate regarding this phenomenon: one hypothesis being that the heads selected by DSP may adapt to suit the convolution constraints, even if they are not met before fine-tuning.

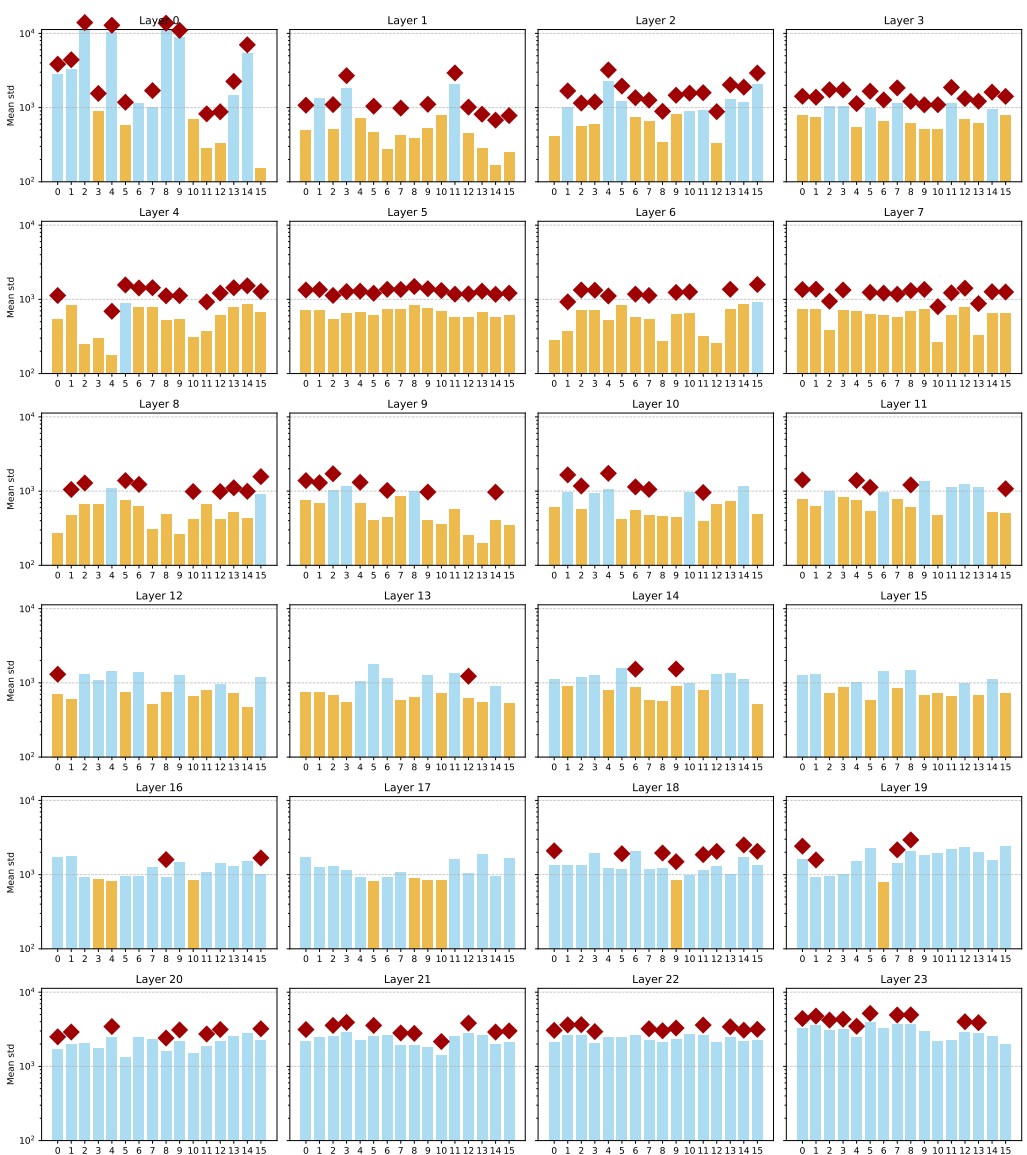

Figure A.4: Values of $\Sigma_h$ for each individual head in Dino-V2 ViT-L, distribution after finetuning on COCO. Orange, 192 heads selected with the lowest $\Sigma_h$ (i.e, scattered criteria). Red diamond indicates heads selected by DSP.

## A.4 FURTHER PROFILING

In Figure A.5, we observe the speedup trend as a function of the number of replaced blocks for both ensembled and unensembled formulations. In our experiments, we replaced up to 16 out of 24 blocks in the ViT-L backbone, achieving over a 20% speedup. In the future, an improved fine-tuning strategy could push the performance boundary, allowing the replacement of more blocks.

### A.4.1 ADDITIONAL HARDWARE

Below, we propose to validate the computational benefits of the proposed method on platforms other than the Nvidia Jetson Orin. The platforms chosen are a second Nvidia board, the least powerful Nvidia Jetson Nano

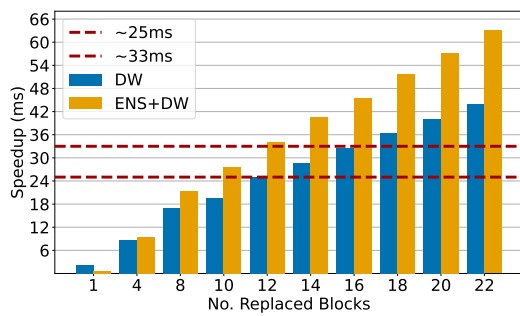

Figure A.5: Number of replaced blocks (blockwise) versus speedup for Depthwise formulations with and without ensembling. Results obtained on ViT-L, resolution $336 \times 336$.

(not to be confused with ORIN Nano), the HAILO-8 AI Accelerator, and CPU platforms ARM Cortex A53 (Mobile CPU) and Intel-Core i7-11700K (Desktop CPU).

Table A.3: Head-level profiling on different hardware platforms.

| Heads | Mode | Orin (ms) | Nano (ms) | HAILO8 (ms) | A53 (ms) | i7 (ms) | Params (M) | FLOPS (G) |
|---|---|---|---|---|---|---|---|---|
| 12 | MhSA | 0.75 | 8.41 | 1.39 | 11.02 | 334.67 | 1.41 | 2.362 |
| 12 | DW | 0.28 | 2.65 | 0.89 | 0.89 | 78.92 | 0.61 | 1.189 |
| 12 | Ens+DW | 0.09 | 0.28 | 0.30 | 0.30 | 7.27 | 0.05 | 0.1 |
| 16 | MhSA | 0.93 | 13.65 | 2.42 | 19.45 | 593.92 | 2.42 | 4.20 |
| 16 | DW | 0.32 | 4.11 | 1.14 | 4.22 | 143.93 | 1.08 | 2.11 |
| 16 | Ens+DW | 0.06 | 0.32 | 0.42 | 0.27 | 9.74 | 0.07 | 0.13 |

Similarly to Table 1, we benchmark the performance of a single attention block of $n_h$ heads, comparing the full MhSA, our drop-in depthwise formulation (DW), and the depthwise convolution with the addition of head ensembling (Ens + DW). The results obtained confirm the soundness of the proposed approach on a broader set of inference platforms.

### A.4.2 FP16 INFERENCE

TensorRT supports various numerical precisions, but exhaustive comparison is challenging. The compiler optimizes multi-head self-attention (MhSA) via Myelin, an obscure, undocumented backend[2]. In FP16 mode, Myelin automatically replaces MhSA with Flash-AttentionV2 Dao (2024), later referred to as FMhSAV2, a specialized implementation leveraging specific Nvidia GPU features. Since this behavior cannot be disabled, evaluations are restricted to Nvidia Ampere GPUs and newer. In Table A.4 we compare performance in said scenario, showing that the discussed approximations outperform full self-attention even in this challenging scenario.

Table A.4: Head-level inference performance comparison at FP32 and FP16 precision.

| Attention | Inference (ms) | Memory (MB) |
|---|---|---|
| MhSA (FP32) | 3.2 | 47.2 |
| Conv-DW (FP32) | 1.26 | 6.75 |
| Ens+DW (FP32) | 0.215 | 0.288 |
| **FMhSAV2** (FP16) | 1.14 | 5.62 |
| Conv-DW (FP16) | 0.86 | 3.38 |
| Ens+DW (FP16) | 0.271 | 1.27 |

### A.4.3 USAGE OF LARGE LANGUAGE MODELS (LLMS)

The authors specify that the use of LLMs in the development of this work and this manuscript is limited and contingent as support in the writing (spell checking, suggestions on phrasing, and helping with LaTeX constructs).

---

[2] https://github.com/NVIDIA/TensorRT/issues/2576

