# OpenReview forum: "Accelerating Vision Transformers with Drop-in Depthwise Convolution"
_ICLR.cc/2026/Conference — ICLR 2026 Conference Withdrawn Submission_

### Official Review · Reviewer_Hu1v · 2025-10-24

**Soundness:** 2
**Presentation:** 2
**Contribution:** 2
**Rating:** 4
**Confidence:** 4

**Summary:**

In this paper, an acceleration method for ViT in the inference stage is proposed. Specifically, the method use depth-wise convolution layers to replace some of the attention heads to improve the inference efficiency. Moreover, a layer selection criterion is proposed to determine which MhSA blocks should be replaced. The experimental results on COCO (object detection tasks) and ImageNet (image classification tasks) shows that the proposed method may result in performance loss, but improve the inference speed.

**Strengths:**

1. The theoretical derivation of the method is clear. The authors clearly presented the procedure of how to use depth-wise convolution to replace the multi-head self attention.

2. The experiments cover large-scale image classification and object detection tasks, and can, to a large extent, support the argument of the paper.

**Weaknesses:**

1. The proposed method is mainly focuses on the inference stage. It is better to consider if the conv-replaced networks can be trained from scratch and also achieve good performance.

2. As the section of related works mentioned, there are many kinds of methods work in the inference stage to improve the efficiency. It is better to include some of those methods, such as network pruning and token reduction, as baseline of experiments to compare with the proposed method.

3.  Some typos should be fixed. For instance, the title of section 1 ( it should be 'Introduction') is missing.

**Questions:**

My questions are included in the Weaknesses part.

---

### Official Review · Reviewer_JBdg · 2025-10-28

**Soundness:** 2
**Presentation:** 1
**Contribution:** 2
**Rating:** 2
**Confidence:** 4

**Summary:**

This paper addresses the critical challenge of deploying large-scale pretrained Vision Transformers (ViTs) on resource-constrained or edge devices, where high inference costs limit applicability. The core of the proposed solution is an efficient, drop-in replacement strategy that leverages the model's intrinsic characteristics. Building on the premise that some Multi-head Self-Attention (MHSA) heads learn highly localized, static feature patterns, a behavior resembling standard convolution, the authors introduce an efficient depth-wise convolution layer to directly replace a carefully selected subset of these heads. This substitution eliminates the computationally intensive $QK^T$ and Softmax steps for those specific heads, accelerating the process while retaining the pretrained weights. The method includes strategies to identify the ideal candidates for replacement and a minimal fine-tuning procedure to recover downstream performance. Validated across image classification and segmentation tasks on low-power platforms, the approach successfully achieves a significant 17–20% inference speedup with performance degradation.

**Strengths:**

The following are some strengths of this work:

1. The paper addresses the important challenge of deploying large-scale pretrained Vision Transformers on resource-constrained devices, where high inference costs limit applicability.
2. The core solution is an efficient, drop-in replacement strategy that utilizes the observation that some Multi-head Self-Attention heads exhibit a localized, static feature pattern similar to standard convolution. The authors introduce an efficient depth-wise convolution (DWC) layer to replace a carefully selected subset of these heads.
3. This substitution eliminates the complexity associated with the $QK^T$ and Softmax calculations for those heads, accelerating the process while preserving pretrained weights. The method includes strategies to identify suitable replacement candidates and a small fine-tuning procedure to recover downstream performance.
4. Tested across image classification and segmentation tasks on low-power platforms, the approach achieves a 17–20\% inference speedup with minor performance degradation, offering a practical method for accelerating foundational vision models for deployment.

**Weaknesses:**

The following are some concerns i request the authors to address:
1. The use of convolutional kernels to improve efficiency in self-attention block has been proposed in several works [1*, 2*, 3*]. So this idea of using depth-wise convolution is not new. Notably, in [1*] the authors not only use it to disentangle information across attention heads (i.e. they term it as refining attention heads), but show that it converges faster and improves the performance over baseline on different downstream tasks. The use of convolution kernel in [1*] can be replaced exactly in eq 8 of this manuscript. Additionally [2*] introduces a parametric efficient way involving MLP and depth-wsie convolution layer to address high-correlation across MHSA blocks to improve efficiency. So I'm not sure whether the current contribution of the manuscript is different to that of these works and the authors do not cite or discuss them, even though these works are quite recent and popular.
2. Han et al. [4*] shows that the behavior of depth-wise convolution operation resembles local attention, which helps learn translation equivalent representations.  The approximation restricts certain heads to a purely local receptive field $\Delta_k$. Since the core strength of attention (Equation 7) lies in its *global receptive field* ($\Delta_m$), replacing heads with local operations risks severely compromising the model's ability to aggregate long-range dependencies. Can the authors please justify how the attention heads are guaranteed to capture and relay the necessary global information, or provide a detailed analysis of the performance trade-off associated with this loss of global interaction.
3. It is conceptually confusing to present the substitution of the dynamic, data-dependent attention matrix $E_h(X)$ (derived from $QK^T$ and Softmax) with a fixed, learned convolutional kernel $K^h$. Can the authors please justify replacing an adaptive weighting function with a fixed kernel parameter, especially since the core computational cost of full attention is tied to the complexity of generating $E_h(X)$.
4. The work focuses heavily on accelerating standard ViT backbones. It is unclear how this Depth-wise convolution kernels  and the associated head-selection criteria (based on $\Sigma_h$) would generalize to more complex or already-efficient transformer architectures, such as *pyramidical architectures* (e.g., PVT) or models using *local windowed attention* (e.g., Swin Transformer). Since many advanced vision models already employ local attention to reduce complexity, can the  authors please discuss the feasibility and the computational benefit (if any) of applying DwC replacement in these contexts, specifically addressing how the *selection criteria* adapt to non-global attention matrices.
5. Im not fully convinced with the performance of the proposed idea. From Table 2, we observe that although there is an improvement in speedup, the drop in dense performance of (2\%) and classification drop of 1.6\%, is huge considering other methods [2*] that shows improvement of 0.8\% to 3\% over the baseline on dense tasks while showing speedup. Additionally, the authors do not compare against any of token sampling methods [5*, 6*] or ToMe [Bolya et al.] or efficient attention methods [1*, 2*, 3*]. So its unclear how the current method performs with respect to prior works on improving efficiency of vision transformers.
6. I would request the authors to please look into related works section of [2*] to refer and discuss all prior works. The use of hybrid architectures to improve efficiency is also an interesting like of research that needs to be discussed.

[1*] Zhou et al., Refiner: Refining Self-attention for Vision Transformers
[2*] Venkataramanan et al., Skip-Attention: Improving Vision Transformers by Paying Less Attention, ICLR 2024
[3*] Zhang et al., Depth-Wise Convolutions in Vision Transformers for Efficient Training on Small Datasets
[4*] Han et al., On the connection between local attention and dynamic depth-wise convolution, ICLR 2022
[5*] Fayyaz et al., Adaptive token sampling for efficient vision transformers, ECCV 2022
[6*] Tang et al., Patch slimming for efficient vision transformers, CVPR 2022

**Questions:**

1. In L 146, the authors mention that DINOv2 prepend [CLS] token in vision transformers in not true. What i mean is that it is not just DINOv2 who prepend the [CLS] token, but it started from the original ViT [Dosovitskiy et al.]. Also I dont understand why the authors ignore [CLS] in their work? How do the authors evaluate for classification?

---

### Official Review · Reviewer_bvqp · 2025-11-01

**Soundness:** 2
**Presentation:** 2
**Contribution:** 2
**Rating:** 2
**Confidence:** 4

**Summary:**

This paper aims to replace the attention heads in pre-trained vision transformers with an efficient depthwise convolution-based layer to facilitate the efficiency. Specifically, the authors propose simple strategies to identify which heads can be replaced and introduce a fine-tuning procedure that recovers downstream task performance. Empirical results on image classification and segmentation tasks demonstrate the efficacy of the proposed method.

**Strengths:**

1. The idea of replacing attention heads with depthwise convolution-based layer seems interesting and empirical results show that doing so could bring speedup the inference process to some extend without significant performance degradation.
2. It is appreciated that the authors have conduct experiments on different vision backbones and various tasks, including ImageNet classification and semantic segmentation.
3. The writing is clear and the paper is easy to follow. The overall structure is well organized and the idea is presented in a coherent manner.

**Weaknesses:**

1. There are limited model compression baselines shown in the paper. The proposed method can be regarded as a model compression method and a practical question is: does the proposed method has advantages over existing model compression techniques, like pruning, distillation and quantization? It seems that the actual speedup brought by the method is sort of limited, i.e., 20% or less, compared to existing model compression techniques.
2. The authors said that they will perform fine-tuning on the target task with regular MhSA heads at first, and the reviewer wonders is there a particular reason for this? More importantly, will the baseline methods (MhSA) be fine-tuned with similar training costs or smaller?
3. While the authors have evaluated the proposed method on ViT-Base and ViT-Large level architectures, it would be better to examine the effectiveness of the method on smaller variants as well, such as ViT-Small and ViT-Tiny, which may be more important for practical applications.

**Questions:**

See weakness above.

---

### Official Review · Reviewer_tCsA · 2025-11-01

**Soundness:** 1
**Presentation:** 2
**Contribution:** 1
**Rating:** 2
**Confidence:** 4

**Summary:**

Current vision foundation models are hard to deploy on resource-constrained devices. To address this, the authors propose accelerating large-scale pretrained ViTs by replacing attention heads with depthwise convolutions. Extensive experiments demonstrate the effectiveness of the proposed method.

**Strengths:**

1. Extensive experiments show the effectiveness of proposed method.

**Weaknesses:**

1. Use “MHSA” instead of “MhSA.”

2. The authors claim in line 196: “We approximate attention by assuming that some heads can be replaced by input-independent kernels restricted to a local neighborhood.” While all attention heads are input-dependent and exhibit global interactions, is there any theoretical justification for approximating them as (1) input-independent and (2) limited to a local neighborhood? How are these heads identified?

3. The authors claim in line 246 that “Given a target of ph heads to approximate with convolution, selection can be either scattered, replacing arbitrary heads across the model, or blockwise, replacing all nh heads within pb MhSA blocks.” If the authors can identify heads that can be approximated by convolution, why should blockwise replacement work in theory? I assume “blockwise” means replacing the entire attention block with convolution.

4. I do not see why the standard deviation can be a simple and effective proxy for identifying convolution-like heads.

5. It is unclear why the authors consider $N_h=n_h*n_b$. Heads in different blocks may have different functionalities. Although they have the same index, they are not in one-to-one correspondence.

6. How do the authors obtain the 600 GB memory estimate in line 267? Does it account for batch size, dimension, model depth, etc.?

7. What exactly is being summed in Eq. 15? is it the sum of attention-score standard deviations across all heads?

8. This evaluation seems unreasonable; we would expect an assessment of the zero-shot ability of foundation models.

**Questions:**

1. The authors could present the idea more clearly; introducing numerous new terms does not increase the theoretical contribution given the simplicity of the idea.

2. Please follow common mathematical practice. For example, $\bar{w}$ does not usually denote a top-k operation. The abundance of such arbitrary definitions makes the paper hard to follow.

3. Suppose we randomly remove n blocks from a pretrained large ViT and then fine-tune the model, how would the performance compare to the proposed method?

4. In general, combining convolutions and attention has been extensively studied. This paper does not introduce particularly novel insights.

---

### Note · Authors · 2025-11-27

**Comment:**

We sincerely thank the reviewers for their thoughtful feedback. We have decided to withdraw our submission in order to further revise and strengthen the work.

**Withdrawal Confirmation:**

I have read and agree with the venue's withdrawal policy on behalf of myself and my co-authors.